# Subsurface Insight: Spatio-Temporal Embeddings for Hydrogeological Time Series Analysis

Anonymous Full Paper
Submission 42

## Abstract

The generation of terabytes of hydrogeological sensor data each year has driven a growing demand for hydrological data analysis to aid in water resource prediction and management. However, these data present complex spatio-temporal dependencies, especially for large-scale data. Traditional statistical methods like PCA often fail to capture nonlinear temporal patterns, and existing deep learning approaches do not effectively focus on integrating spatial context. This paper introduces a deep spatio-temporal autoencoder architecture to learn embeddings from decades of French groundwater level data. By using contrastive learning, we combine time series and their geographical coordinates to generate low-dimensional embeddings. We demonstrate that these embedding vectors are highly effective for downstream tasks, unsupervised clustering, compared with traditional methods. Crucially, our approach achieves a **Normalized Mutual Information (NMI) score exceeding 0.55** against an expert-labeled ground truth, confirming that the learned representations capture physically meaningful subsurface characteristics.

## 1  Introduction

Understanding groundwater level is essential for managing groundwater resources, supporting forecasting and assessing geological hazards. Nowadays, modern sensors generate vast amounts of continuous data, but extracting meaningful patterns is always challenging due to noise, non-linearities, and intricate dependencies across multiple temporal and spatial scales. While traditional statistical methods like PCA are limited by their linearity and inability to handle sequential data, or models like ARIMA do not work well for Long-term time series data. To address these limitations, we propose a deep learning framework for spatio-temporal representations from hydrogeological data. With the contributions: (1) We introduce a autoencoder architecture using CNN and MLP-Mixers [1] for long-range dependency, (2) We demonstrate that adding spatial coordinates significantly improves the quality of embedding vectors (3) we also demonstrate that the embedding vectors from our model capture a classification structure that strongly aligns with expert-defined labels.

## 2  Methodology

### 2.1  Data and Preprocessing

We use a large-scale dataset from France since 1960 [2]. Let the initial raw dataset be denoted as $\mathcal{D}_{\text{raw}} = \{(X_i, C_i)\}_{i=1}^N$, where $X_i$ is the raw time series from sensor $i$, and $C_i$ represents its geographical coordinates. First, missing values are filled using surrounding values. However, to avoid affecting the distribution of the data when sampling, any time series with a continuous gap exceeding 30 days is segmented at the missing. This process each original series $X_i$ into a set of one or more continuous sub-series, or segments, $\{S_{i,1}, S_{i,2}, ..., S_{i,k_i}\}$, where $k_i \geq 1$. Next, each outlier in segment $S_{i,j}$ is removed using the interquartile range (IQR) method, where any data point outside the range is clipped. After that, we apply two-step normalization. First, a Box-Cox power transformation is applied to stabilize variance. Then, we apply Z-score normalization to ensure consistency across the entire data set.

Finally, our dataset is a collection of all normalized segments, each paired with its original sensor's normalized coordinates, denoted as $\{(S'_{i,j}, C'_i)\}$. This structure is specifically designed for our contrastive learning framework, where any two segments originating from the same sensor $i$ (e.g., $S'_{i,j}$ and $S'_{i,l}$) constitute a positive pair.

### 2.2  Model Architecture.

Our model is an autoencoder inspired by TimeMixer [3, 4] and uses a convolutional neural network (CNN) [5]. As illustrated in the figure 1, it processes two inputs: a time series window $W \in R^{L \times 1}$ and its geographical coordinates $C \in R^2$.

The core of the **encoder** consists of two parallel processing paths followed by a fusion layer. The time series path first processes the input window $W$ through a series of 1D **CNN blocks (A)** to extract a hierarchy of local features. These features are then combined by several **Mixer Blocks** to capture long-range connections. Concurrently, the **spatial path** processes the geographical coordinates $C$ by an MLP to produce a location embedding. Crucially, the outputs from both the temporal path and the spatial path are then **concatenated** and passed through a final **fusion layer** (MLP) to create the latent embedding $e$. The latent embedding $e$ serves as the

input for two downstream heads, each corresponding to a distinct training objective. First, it is passed to the **decoder**, which reconstructs the original time series $\hat{W}$ using a U-Net-like architecture to ensure high-fidelity output [6]. Second, $e$ is fed through a **projection head** for the contrastive learning task.

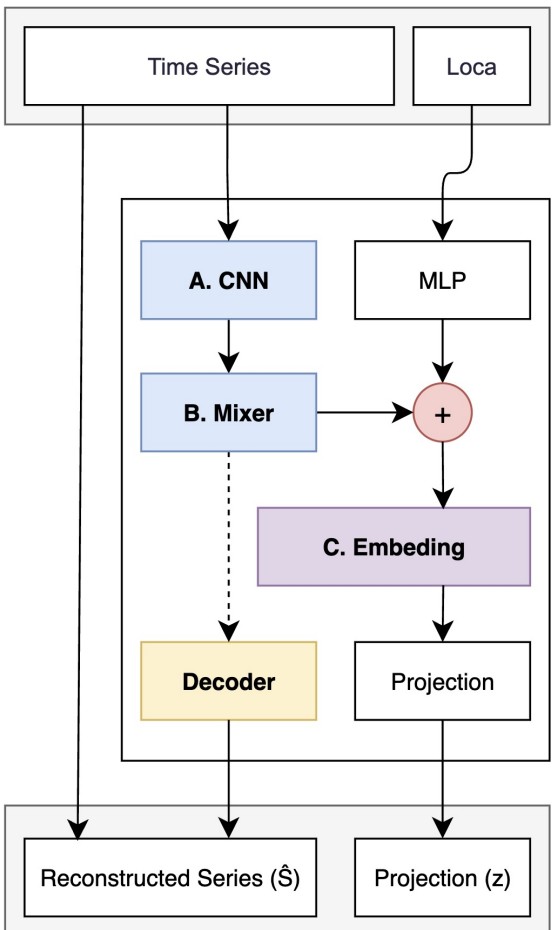

**Figure 1.** Our final architecture, henceforth referred to as 'Our Model', consists of a CNN branch for local feature extraction and an MLP-Mixer branch for long-range dependencies

**Training Objective**, the model is trained end-to-end to minimize a composite loss function, which combines a reconstruction objective and a contrastive learning objective:

$$\mathcal{L}_{\text{total}} = \mathcal{L}_{\text{contrastive}} + \alpha \mathcal{L}_{\text{recon}}$$

Here, $\mathcal{L}_{\text{recon}}$ is the Mean Squared Error (MSE) between $W$ and $\hat{W}$ and The hyperparameter $\alpha$ is a weighting factor that balances the contribution of the reconstruction loss. For the contrastive task, we pass the embedding $e$ through a projection and compute the NT-Xent loss [7], which encourages similar time series to have closer embeddings in the projection space. After training, only the encoder and fusion modules are used for downstream tasks.

## 3 Results and Analysis

We trained our model on a training set of **8,805 segments**, validating on **2,202 segments**. Models were trained using the Adam optimizer with a learning rate of $10^{-4}$ and early stopping. The evaluation of the embedding quality was done in two stages.

First, to validate our architectural design choices, we conducted an ablation study comparing our full model against several variants using internal **clustering metrics (k=4)**. Our full proposed architecture is denoted as "Our Model". We compare it against a PCA baseline and key architectural variants to justify our design choices. The results are summarized in Table 1. Our full model achieves the highest Silhouette Score and lowest WCSS. Notably, (Our Model (no loca)) significantly decreases performance, confirming the importance of integrating spatial information.

Second, to validate embedding vectors quality in the real-world, we use a small dataset, experts classified it into "IG" sectors based on geological, hydrogeological, topographical, and administrative criteria, which were used as reference labels in the BSN study [8] to evaluate clustering performance, we clustered the embedding vectors into **66 groups**, corresponding to the number of labeled. This comparison yielded a **Normalized Mutual Information (NMI) score exceeding 0.55**, which shows a strong correspondence of our clustering with the expert classification, and indicates that the model has learned to capture hydrologically significant features.

**Table 1.** Comparison of clustering performance. "Our Model (CNN+Mixer)" is our full proposed architecture. Bold values indicate the best result for each metric. For WCSS and DBI, lower scores are better.

| Model | Sil | WCSS | DBI |
|---|---|---|---|
| **Our Model** | **0,58** | **36.279** | 0,60 |
| Our Model (no loca) | 0,50 | 175.329 | 0,65 |
| PCA | 0.53 | 447.094 | **0,55** |
| CNN+Att | 0.54 | 171.369 | 0,62 |
| CNN+Mixer+Att | 0.35 | 265.529 | 1,19 |

## 4 Conclusion

We have presented an architecture effectively captures complex patterns, and we also demonstrate that incorporating geographical information is crucial for learning high-quality representations. The embedding vectors work well in unsupervised clustering, proven effective on real data. Future work will explore applying these embeddings to downstream tasks such as forecasting and anomaly detection.

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
