# OpenReview forum: "Subsurface Insight: Spatio-Temporal Embeddings for Hydrogeological Time Series Analysis"
_NLDL.org/2026/Abstracts_Track — NLDL 2026 Abstracts_

### Official Review · Reviewer_oGig · 2025-10-24

**Soundness:** 4
**Correctness:** 3
**Rating:** 5
**Confidence:** 4

**Summary:**

The abstract presents a novel approach to handle spacio-temporal dependencies in large-scale hydrogeological data using a multimodal deep learning model. The input, in form of a time-serie sequence and its corresponding spatial coordinates, are processed by modality specific featurizers and fused using an mlp to create a common embedding. The model is trained in a self-supervised way, combining a reconstructive and contrastive loss.

**Strengths:**

Provides some results and comparisons to other approaches and a PCA baseline. Introduces a modern, advanced deep learning methodology into a new usecase scenario.

**Weaknesses:**

A more thorough description of the input data would be welcomed, but sufficient for this format. A few typos present.

---

### Official Review · Reviewer_fm81 · 2025-10-28

**Soundness:** 2
**Correctness:** 2
**Rating:** 4
**Confidence:** 4

**Summary:**

The abstract proposes a reconstruction based self-supervised learning procedure to embed time series segments alongside their geographical coordinates.

**Strengths:**

Time series clustering is a challenging task, making the proposal of new methodology and its application in real life data highly relevant. The authors report a number of experiments, including some ablation study of their method.

**Weaknesses:**

The abstract submission was supposed to be single-blind including author names and affiliations, which gave other authors who adhered to the guidelines less space to write their submission on two pages.

The dataset description does not convey to me what this time series really represents. Hydrogeological data, yes, but is it ground water height this sensor is measuring? Water pressure? Flow rate?

The writing could be improved. Phrases like "[...] any time series [...] is segmented at the missing" is incorrect and adds ambiguity to what is actually done in the work, similarly sentences like "This process each original series $X_i$ into a set [...]" are missing a verb and are hence left for interpretation of the reader.

It is unclear how a time series window W as introduced on line 78 relates to the time series $X_i$ or its segments as used in the paper previously.

Some abbreviations such as WCSS or DBI are not introduced. These happen to be two of the three evaluation measures and I don't know what the measures stand for, so it is impossible to interpret the results.

It is unclear to me how the evaluation with a small annotated dataset was used. Why was normalized mutual information used, not label accuracy interpreting the found labels as clusters or measuring the homogeneity of labels in clusters for the datapoints which had labels?

Some references are missing in the text, e.g. to ARIMA.

---

### Official Review · Reviewer_H9Jh · 2025-11-02

**Soundness:** 3
**Correctness:** 3
**Rating:** 4
**Confidence:** 3

**Summary:**

The abstract addresses the issue of creating spatio-temporal embeddings for hydrogeological time series. These data present complex spatio-temporal dependencies and to incorporate this a deep spatio-temporal autoencoder is suggested where contrastive learning is used, combining time series and their geographical coordinates to generate low-dimensional embeddings. The quality of the embeddings is tested through unsupervised clustering and compared to other approaches and to expert classifications. The results indicate that the addition of spatial coordinates specifically improves the quality.

**Strengths:**

The idea of the combination of time series and their geographical coordinates in the embedding can have some interest.

**Weaknesses:**

The abstract has some limitations when it comes to language quality.
There is a detailed description of the preprocessing, but it would be good also with more details on the original data and sensor.
Some more comments on the results could also be good where the “our model” (CNN+Mixer), seems to have much better performance than (CNN+Mixer+Att), but slightly lower performance than (CNN+Att).

---

### Decision · Program_Chairs · 2025-11-05

**Decision:**

Accept

**Comment:**

The abstract is of interest to the community and should be presented at the conference.